# Explaining the Lack of Mesh Convergence of Inviscid Adjoint Solutions near Solid Walls for Subcritical Flows

Carlos Lozano *[ID] and Jorge Ponsin

Computational Aerodynamics Group, National Institute of Aerospace Technology (INTA), Carretera de Ajalvir, Km 4, 28850 Torrejón de Ardoz, Spain; ponsinj@inta.es
* Correspondence: lozanorc@inta.es

**Abstract:** Numerical solutions to the adjoint Euler equations have been found to diverge with mesh refinement near walls for a variety of flow conditions and geometry configurations. The issue is reviewed, and an explanation is provided by comparing a numerical incompressible adjoint solution with an analytic adjoint solution, showing that the anomaly observed in numerical computations is caused by a divergence of the analytic solution at the wall. The singularity causing this divergence is of the same type as the well-known singularity along the incoming stagnation streamline, and both originate at the adjoint singularity at the trailing edge. The argument is extended to cover the fully compressible case, in subcritical flow conditions, by presenting an analytic solution that follows the same structure as the incompressible one.

**Keywords:** adjoint Euler equations; analytic adjoint solution; wall singularity; mesh dependence





## 1. Introduction

In a series of recent papers [1–4], it has been established that certain numerical adjoint solutions to the two and three-dimensional Euler equations have values at and near the surface of wings and airfoils that depend strongly on the mesh density, and which do not converge as the mesh is refined. This phenomenon has been observed for lift-based adjoint solutions for any subcritical or transonic flow condition and also for incompressible flow, while for drag-based adjoint solutions, it has only been observed in transonic rotational flows.

The problem seems to be rather generic, as it has been found in solutions obtained with continuous and discrete adjoint schemes and with different solvers. Increasing the numerical dissipation with mesh refinement does not qualitatively change the behavior, although the actual value of the adjoint variables at the wall strongly depends on the level of numerical dissipation. It was conjectured in [1] that this behavior is likely caused by the adjoint singularity at the sharp trailing edge, although an understanding of the actual mechanism was lacking. It was subsequently pointed out that the anomaly is also correlated with the adjoint singularity along the incoming stagnation streamline predicted by Giles and Pierce [5] and that it also appears in flows past blunt bodies without sharp trailing edges. Finally, recent evidence [6] involving point perturbations shows that there might actually be an adjoint singularity along the wall of the same origin as the one along the incoming stagnation streamline. Unfortunately, exact adjoint solutions for the Euler equations in two and three dimensions were not available until recently. In [7], an analytic adjoint solution for the incompressible 2D Euler equations was obtained using the Green's function approach [5,8]. It turns out that the lift-based adjoint variables diverge along the incoming stagnation streamline, as expected, but also at the wall. In the remainder of the paper, we will swiftly review the mesh divergence problem and the derivation of the analytic adjoint solution. A detailed comparison of the analytic solution with a numerical adjoint solution exhibiting the mesh divergence problem demonstrates that the behavior observed in numerical solutions corresponds to the singularity of the analytic solution.

An analytic solution for the fully compressible case in subcritical flow conditions is also presented. The solution is given in terms of two unknown functions that encode the effect of perturbations to the Kutta condition and that obey two differential equations along the streamlines. No closed-form solution is available in this case, but the structure of the solution closely follows that of the incompressible case. This fact, together with the observation that the structure of numerical subcritical adjoint solutions is very similar to the incompressible case, allows us to conjecture that the cause of the divergence is the same in this case as well.

## 2. Review of the Mesh Divergence Problem

To introduce the problem, we examine a fairly simple example: The adjoint solution for inviscid incompressible flow at an angle of attack $\alpha = 0°$ past a symmetrical van de Vooren airfoil given by the conformal transformation [9]

$$z(\zeta) = \frac{(\zeta - R)^k}{(\zeta - \sigma R)^{k-1}} + 1 \tag{1}$$

where $\sigma$ is a thickness parameter and $k$ is related to the trailing-edge angle $\tau$ as $k = 2 - \tau/\pi$. The transformation maps the airfoil in the complex $z$ plane to a circle of radius $R = (1 + \sigma)^{k-1}/2^k$ centered at the origin in the $\zeta$ plane. In this paper, we set $\sigma = 0.0371$ and $k = 86/45$, resulting in an airfoil with 12% thickness and finite trailing edge angle $\tau = 16°$ that closely resembles a NACA0012 airfoil. An exact flow solution for this case can be obtained with conformal mapping techniques (see [9] and Section 3.1 below) and is shown in Figure 1, along with the geometry of the airfoil.

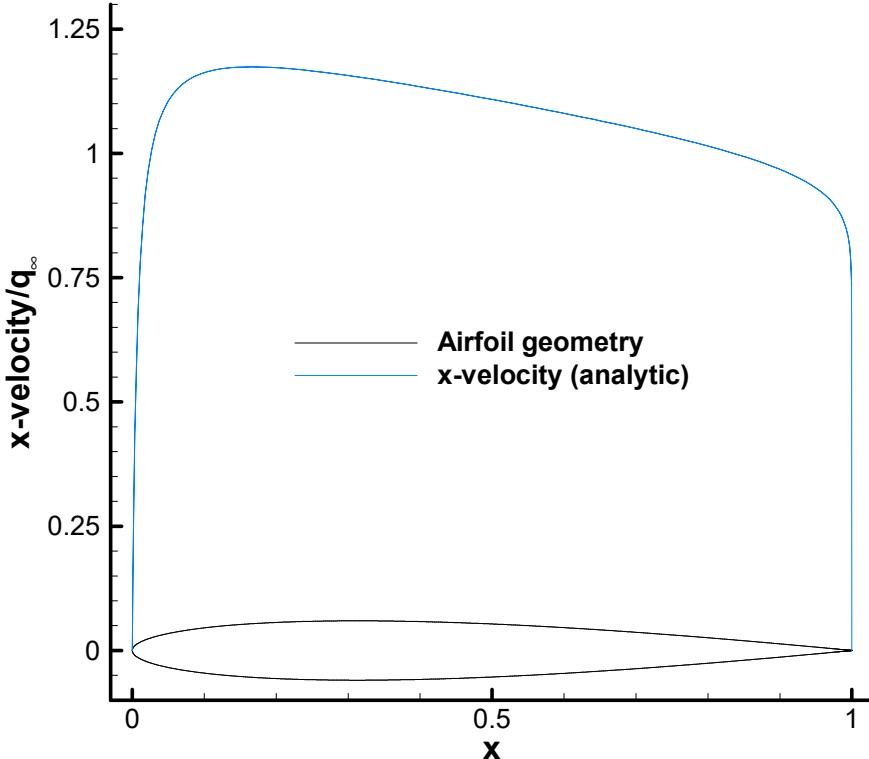

**Figure 1.** Analytic $x$-velocity for incompressible, inviscid flow at $\alpha = 0°$ past a van de Vooren airfoil with trailing–edge angle $\tau = 16°$ and 12% thickness computed with conformal transformation techniques.

Let us next consider the lift or drag-based adjoint problem for this flow. The adjoint problem is aimed at computing the sensitivity of a functional of the flow or cost function

with respect to perturbations of the flow. (For a more thorough introduction to the adjoint method, see, for example, Refs. [10–12]). In the present case, the cost function is the aerodynamic lift or drag coefficient computed as

$$\int_S C_p \left( \vec{n}_S \cdot \vec{d} \right) ds \tag{2}$$

where $S$ denotes the wall boundary, $C_p = (p - p_\infty)/c_\infty$ is the non-dimensional pressure coefficient, $p$ is pressure, $\vec{n}_S$ is the outward pointing unit normal vector at the wall, $\vec{d} = (\cos\alpha, \sin\alpha)$ for drag and $\vec{d} = (-\sin\alpha, \cos\alpha)$ for lift, respectively, where $\alpha$ is the angle of attack, $c_\infty = \rho q_\infty^2 \updownarrow /2$ is a normalization constant, $\rho$ is the (constant) density, $q_\infty^2 = u_\infty^2 + v_\infty^2$ is the fluid velocity at the farfield and $\updownarrow$ is a reference length scale (typically the airfoil chord length).

The adjoint state $\psi^T = (\psi_1, \psi_x, \psi_y)$ obeys the adjoint Euler equation

$$\nabla \psi^T \cdot \vec{F}_U = 0 \tag{3}$$

with adjoint wall boundary condition

$$(\psi_x, \psi_y) \cdot \vec{n}_S = c_\infty^{-1} \vec{d} \cdot \vec{n}_S \tag{4}$$

and far-field b.c.

$$\psi^T (\vec{F}_U \cdot \vec{n}_{S\infty}) \delta U = 0. \tag{5}$$

Here, $\vec{F}_U = \partial(\rho\vec{v}, \rho\vec{v}u + p\hat{x}, \rho\vec{v}v + p\hat{y})^T / \partial(p, \rho u, \rho v)$ is the (incompressible) flux Jacobian, $\vec{v} = (u, v)$ is the fluid velocity and $\delta U = (\delta p, \delta(\rho u), \delta(\rho v))^T$ is a linearized flow perturbation. This case should be straightforward to solve numerically, but it turns out to yield unexpected results, as we will see momentarily.

The drag and lift-based adjoint solutions corresponding to this case have been computed with the SU2 incompressible solver [13], which will be used for numerical testing in the remainder of the paper on a sequence of five progressively refined unstructured triangular meshes (labeled as meshes 1–5). SU2 solves the incompressible Euler equations with artificial compressibility [14] using a cell-vertex finite volume central difference scheme with JST dissipation [15] described in detail in [16].

The computational meshes are obtained from the basic triangular Euler mesh shown in Figure 2 by uniform refinement. The initial mesh (mesh 1) has 400 nodes on the airfoil profile and 6211 nodes, and 11,970 triangular elements throughout the flowfield, with the far-field placed at 100 chord lengths. At each refinement stage, every edge is bisected, and the resulting nodes are joined to form new triangles. In order to preserve the surface of the airfoil, a Bézier-spline surface reconstruction on the basis of the previous mesh is performed at each stage. The refined meshes 2–5 have $2.44 \times 10^4$, $9.67 \times 10^4$, $3.85 \times 10^5$, and $1.54 \times 10^6$ nodes, respectively, with the final mesh having 6400 nodes on the airfoil profile and $3.06 \times 10^6$ triangular elements throughout the flowfield.

Figure 3 shows the flow variables along the airfoil profile for the sequence of meshes described above. The flow variables are quite accurate and converge with mesh refinement. The adjoint solution, on the other hand, shows a strikingly different behavior. Plotting the adjoint variables on the surface of the airfoil across the different mesh levels, one would expect to see at most a singularity at the trailing edge [17], with the solution along the remainder of the profile remaining stable or progressively converging over successive mesh levels. This is not what is observed, though. As can be seen in Figure 4, the drag-based adjoint solution (left) behaves smoothly even at the trailing edge and converges with mesh refinement, while the lift-based adjoint solution (right) diverges at the trailing edge on any given mesh and its value along the remainder of the airfoil grows continually as the mesh density increases. In Figure 4 and in the remainder of the paper, adjoint variables are non-

dimensional. Dimensions can be restored by multiplying the variables by the appropriate powers of $\rho_\infty$ and $\left|\vec{v}_\infty\right|$, namely $\psi_1 = \overline{\psi}_1 / \left(\rho_\infty \left|\vec{v}_\infty\right|\right)$, $\psi_{x,y} = \overline{\psi}_{x,y} / \left(\rho_\infty \left|\vec{v}_\infty\right|^2\right)$, where overbars denote non-dimensional variables.

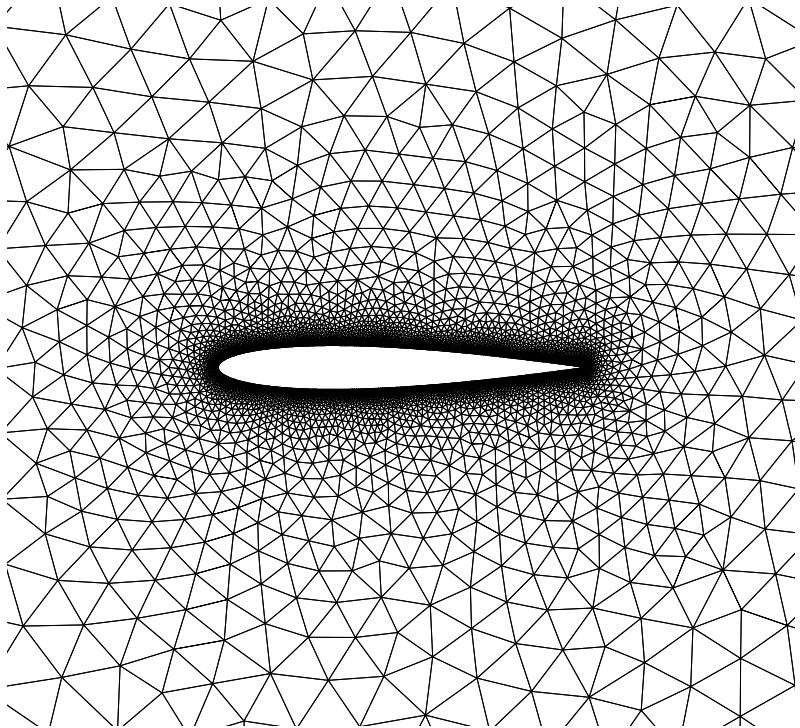

**Figure 2.** Close-up of the baseline computational mesh.

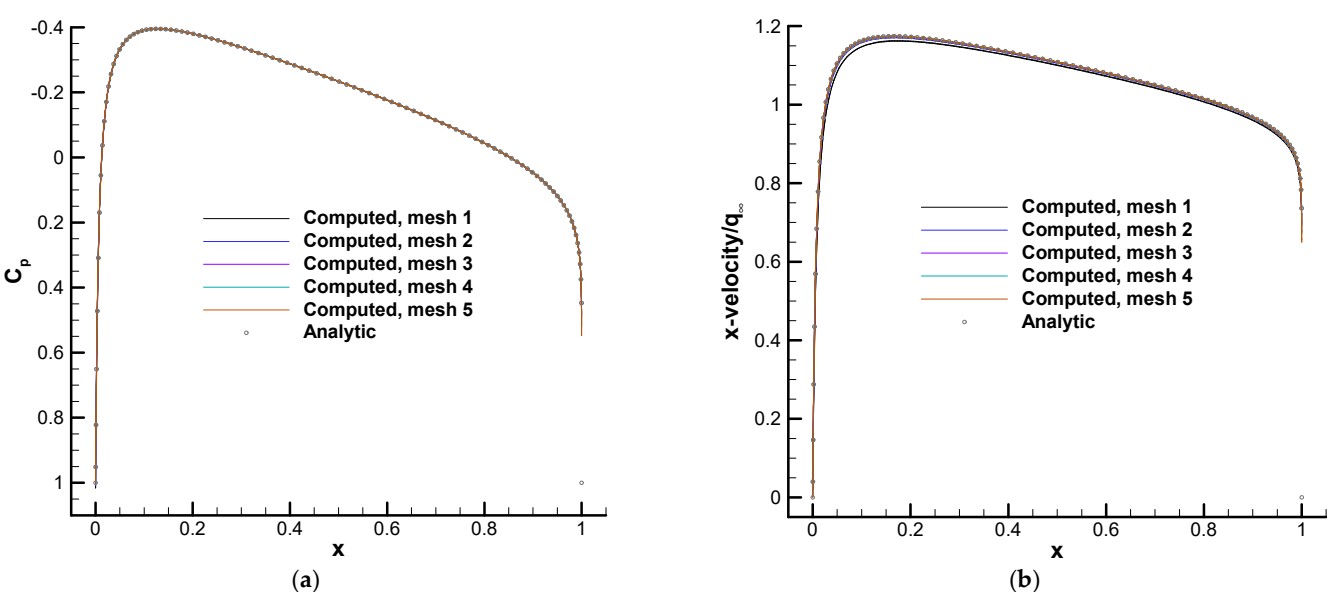

**Figure 3.** Inviscid incompressible flow past a van de Vooren airfoil profile at $\alpha = 0°$ with trailing–edge angle $\tau = 16°$ and 12% thickness. Pressure (**a**) and *x*-velocity (**b**) on the airfoil profile computed with the SU2 solver on a sequence of 5 progressively refined unstructured triangular meshes.

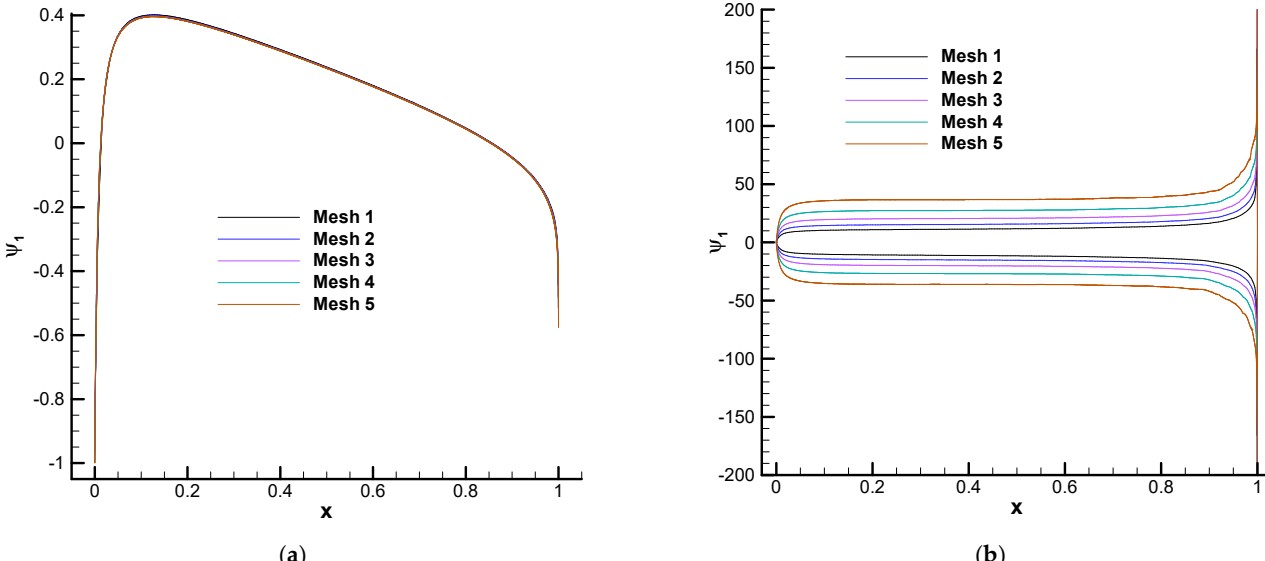

**Figure 4.** Drag (**a**) and lift (**b**) -based inviscid, incompressible adjoint solution on a van de Vooren airfoil profile at $\alpha = 0°$ with trailing–edge angle $\tau = 16°$ and 12% thickness computed with the SU2 solver on 5 progressively refined unstructured triangular meshes.

This anomalous behavior was originally found in [18] for the drag-based adjoint solution corresponding to two-dimensional, transonic inviscid flow past a NACA0012 airfoil at Mach number $M = 0.8$ and angle of attack $\alpha = 1.25°$. In order to study the problem in more depth, the following tests were made in Refs. [1–4]:

1. Viscous cases were investigated to determine if the anomaly is limited to inviscid cases. It is.
2. The effect of cost function and flow regime was tested, with the following results:

- For supersonic flow, neither lift nor drag-based adjoint solutions show this behavior.
- For transonic, subsonic, and incompressible flow, lift-based adjoint solutions are always affected, while drag-based solutions are only affected for transonic rotational flows (such as, for example, shocked flow past a symmetric airfoil with non-zero angle of attack).
- The adjoint state based on the far-field entropy flux $\int_{S_\infty} s\vec{v} \cdot \hat{n} ds$ shows the same behavior as the near-field drag ($s$ is the entropy). This output function is not based on near-field computations and, accordingly, the adjoint wall boundary condition is simply $(\psi_x, \psi_y) \cdot \vec{n}_S = 0$ in this case [19,20].

3. Inviscid three-dimensional cases were tested, and the same behavior was found.
4. The behavior of the adjoint wall b.c. (4) with mesh refinement was investigated. It turned out to be well obeyed across mesh levels except in the immediate vicinity of the trailing edge.
5. Given the anomalous behavior, it was mandatory to test whether adjoint-based sensitivity derivatives are affected. They are not. In fact, they are actually quite accurate and fairly stable across mesh levels. This is extremely important, as one of the hallmarks of continuous adjoint methods is the possibility to compute the sensitivities using only the flow and adjoint solutions on the (wall) boundary.
6. The problem was originally discovered with DLR's Tau code [21], which uses an unstructured, cell-vertex, finite-volume solver, and appeared in both the continuous and discrete adjoint solvers with upwind (Roe-type) and central schemes with JST artificial dissipation. However, similar results have been obtained with the SU2 code, ONERA's structured, cell-centered ELSA code [6], and Imperial College's Nektar++ code [22].

7. Originally, the anomaly was observed in airfoils with non-zero trailing-edge angle. In order to determine the effect of the trailing edge geometry, several different configurations, including blunt and cusped trailing edges, were tested. The anomaly was observed in all cases and also in blunt bodies such as circles and ellipses.

8. The effect of the far-field distance, resolution, and the adjoint far-field b.c. was tested, but no significant influence could be established.

9. The adjoint solutions were examined in order to establish whether the anomaly is related to any flow or adjoint singularity. It was found that the anomaly is always accompanied by the presence of an adjoint singularity at the trailing edge or rear stagnation point but also along the incoming stagnation streamline. Conversely, when such singularities are absent, the adjoint solution converges with mesh refinement.

10. The effect of numerical dissipation was tested by using a central scheme with JST artificial dissipation. On the one hand, mesh convergence studies were repeated with dissipation levels increasing with mesh size ($\varepsilon_2 \sim N^{\frac{1}{2D}}$, where $\varepsilon_2$ is the second dissipation coefficient, $N$ is the number of grid nodes and $D$ is the number of spatial dimensions) without significant qualitative changes in the behavior. On the other hand, the actual value of the adjoint solution at the wall on a given mesh was found to depend strongly on the dissipation level in such a way that reducing the dissipation increases the value of the adjoint solution, mimicking the effect of mesh refinement. A similar behavior of the values of the adjoint variables in the near-wall cells in a cell-centered solver has been pointed out in [6].

11. In addition, it was shown in [6] (see also [4]) that linear perturbations to lift or drag caused by numerical solutions containing point singularities corresponding to stagnation pressure perturbations appear to diverge towards the wall, while other point perturbations (mass, normal force or enthalpy, using the nomenclature of [5]) do not. Since point perturbations are closely related to the adjoint state, this result could indicate the presence of a singularity of the adjoint variables at the wall.

## 3. Analytic Adjoint Solution for Incompressible Flow

It was conjectured in [1] that the anomaly is a numerical effect triggered by the adjoint singularity at the trailing edge, but no precise explanation of the actual mechanism responsible for this behavior was given. Local mesh dependence near adjoint singularities is to be expected, but mesh dependence across the entire wall is puzzling unless one is willing to admit the presence of a singularity at the wall. This possibility was, however, erroneously ruled out in [1] based on the lack of positive evidence in the analysis carried out in [5] of the analytic properties of 2D adjoint solutions, which did not give any hint of an adjoint singularity at the wall (but did not preclude it either). Furthermore, even mesh-diverging numerical adjoint variables respect fairly well the wall b.c. and produce well-defined, mesh-converging adjoint-based sensitivity derivatives computed with wall data alone, all of which seemed to rule out a possible adjoint wall singularity. On the other hand, the results presented in [6] with point perturbations give a clue in the precise opposite direction. It appears that the only possibility for further progress is to examine an actual analytic adjoint solution. There is a systematic procedure to build analytic adjoint solutions based on the Green's function approach [8], which is based on the observation that the adjoint variables at a particular point correspond to the cost function evaluated using the Green's function for the same point. By identifying suitable point perturbations whose effect on the cost function is computable, it is possible to obtain the corresponding adjoint solution. This approach was used in [8] to obtain analytic adjoint solutions for the quasi-one-dimensional Euler equations. In the 2D case, the procedure was outlined for the compressible case in [5] and used in [7] to obtain closed-form drag and lift-based analytic adjoint solutions for two-dimensional inviscid incompressible flows around airfoils. The construction for the incompressible case involves three linearly independent Green's functions as many as there are flow equations or adjoint variables. The Green's functions $\delta U^{(j)}(\vec{x}, \vec{\xi})$ are the

linearized response to singular point perturbations with sources $f^{(j)}(\vec{\xi})\delta(\vec{x} - \vec{\xi})$, $j = 1$, ..., 3, and obey the linearized equations

$$\nabla \cdot (\vec{F}_U \delta U^{(j)}(\vec{x}, \vec{\xi})) = f^{(j)}(\vec{\xi})\delta(\vec{x} - \vec{\xi}) \tag{6}$$

where $\delta(\vec{x} - \vec{\xi})$ is the Dirac delta function. By definition of the adjoint state, the effect of $\delta U^{(j)}(\vec{x}, \vec{\xi})$ on the cost function can be computed as

$$\delta I^{(j)}(\vec{\xi}) = \psi^T(\vec{\xi}) f^{(j)}(\vec{\xi}) \tag{7}$$

Conversely, Equation (7) can be used to compute the adjoint variables in terms of linearized functionals $\delta I^{(j)}$. If $f_k^{(j)} = \delta_{kj}$, then $\psi(\vec{\xi}) = (\delta I^{(1)}(\vec{\xi}), \delta I^{(2)}(\vec{\xi}), \delta I^{(3)}(\vec{\xi}))^T$, i.e., the $i$th adjoint variable is equal to the value of the objective function for the $i$th Green's function. For more general source vectors, if the linearized functionals $\delta I^{(j)}$ are known, the adjoint solution can be obtained as

$$\psi^T(\vec{\xi}) = \left(\delta I^{(1)}, \delta I^{(2)}, \delta I^{(2)}\right) \cdot \left(f^{(1)}\middle|f^{(2)}\middle|f^{(3)}\right)^{-1} \tag{8}$$

where $\left(f^{(1)}\middle|f^{(2)}\middle|f^{(3)}\right)$ is a matrix whose columns are the vectors $f^{(j)}(\vec{\xi})$. The basic idea of the approach is then to select the flow perturbations, identify the source terms $f^{(j)}$, evaluate $\delta I^{(j)}$ and finally obtain $\psi$ from (8).

In what follows, we will explain how to obtain the analytic adjoint solution for the van de Vooren airfoil introduced in Section 2, referring the interested reader to [7] for further details.

### 3.1. Analytic Flow Solution

We start by deriving the analytic flow solution corresponding to the case of inviscid, incompressible flow past the airfoil defined by the conformal mapping (1). The map transforms a circle of radius $R = (1 + \sigma)^{k-1}/2^k$ centered at the origin of coordinates in the complex $\zeta = X + iY$ plane to a symmetric airfoil with unit chord and finite trailing edge angle $\tau = \pi(2 - k)$ in the $z = x + iy$ plane. The airfoil geometry is given by

$$z(\theta) = \frac{(Re^{i\theta} - R)^k}{(Re^{i\theta} - \sigma R)^{k-1}} + 1 \tag{9}$$

where $0 \leq \theta \leq 2\pi$ is the polar angle in the $\zeta$-plane, $\sigma = 0.0371$ and $k = 86/45$. The trailing edge of the airfoil is at $\theta = 0$, which corresponds to $z = 1$ and $\zeta_{te} = R$.

Since the flow around the airfoil is irrotational and incompressible, it can be completely described in terms of a complex function, the complex potential, whose real part gives the velocity potential and whose imaginary part gives the stream function. By a well-known feature of conformal mappings (see [23], Chapter 6), the complex potential describing the flow in the airfoil plane is the same as in the circle plane, but the latter is much easier to compute. The complex potential defining a flow with far-field velocity $(q_\infty \cos \alpha, q_\infty \sin \alpha)$ and circulation $\Gamma_0$ around a circle of radius $R$ centered at $\zeta = 0$ in the $\zeta$-plane is [24]

$$\Phi(\zeta) = q_\infty e^{-i\alpha}\zeta + q_\infty e^{i\alpha}\frac{R^2}{\zeta} - \frac{i\Gamma_0}{2\pi}\ln \zeta \tag{10}$$

The Cartesian velocity components $(U, V)$ in the $\zeta$-plane are obtained from the complex derivative of the potential

$$W_\zeta = U - iV = \frac{d\Phi}{d\zeta} = q_\infty e^{-i\alpha} - q_\infty e^{i\alpha} \frac{R^2}{\zeta^2} - \frac{i\Gamma_0}{2\pi} \frac{1}{\zeta} \tag{11}$$

Since the $\zeta$ and $z$ planes are related by a conformal transformation, the potential at a point $z$ on the airfoil plane is simply $F(z) = \Phi(\zeta(z))$, and the corresponding Cartesian velocity components are then

$$u - iv = \frac{dF}{dz} = \frac{d\zeta}{dz} W_\zeta(\zeta(z)) \tag{12}$$

There is a critical point in the conformal mapping (1) at the trailing edge, where $dz/d\zeta = 0$ at $z = 1$. It is clear from Equation (12) that the flow will have a singularity at the trailing edge unless $W_\zeta(\zeta_{te}) = d\Phi/d\zeta|_{\zeta=\zeta_{te}} = 0$. This is the Kutta condition, which physically corresponds to placing a stagnation point in the $\zeta$-plane at $\zeta_{te} = R$. Recalling Equation (11), this can be achieved by giving the hitherto undefined circulation $\Gamma_0$ the value

$$\Gamma_0 = -4\pi q_\infty R \sin\alpha \tag{13}$$

which vanishes for $\alpha = 0$.

### 3.2. Analytic Adjoint Solution

In order to choose the appropriate perturbations, we recall that for 2D compressible flows, Giles and Pierce [5] considered 4 point perturbations: A mass source at fixed enthalpy and stagnation pressure, a point force in the direction normal to the local flow, and point perturbations to the stagnation enthalpy and pressure. The first two correspond to potential flow perturbations (the source and the vortex), while the last two are equivalent if the flow is incompressible. For 2D inviscid, incompressible flow, we choose exactly the same perturbations. If we also restrict ourselves to irrotational base flows, we are left with two known perturbations (the potential source and vortex), whose effect on lift and drag can be computed using complex variable techniques, while the effect of the third one will turn out to be computable in terms of the first two.

- Source and vortex

The relevant source vectors for the first two Green's functions are $f^{(1)}(\vec{\zeta}) = (1, u, v)^T$ for the source and $f^{(2)}(\vec{\zeta}) = (0, v, -u)^T$ for the vortex, respectively. Their linearized contributions to drag and lift, denoted as $\delta D$ and $\delta L$, can be computed with complex variable techniques as

$$\begin{aligned} (\delta D - i\delta L)^{(1)} &= \varepsilon e^{i\alpha}(u - iv - q_\infty e^{-i\alpha}) + i\rho q_\infty \delta\Gamma_0 + O(\varepsilon^2) \\ (\delta D - i\delta L)^{(2)} &= i\varepsilon e^{i\alpha}(u - iv - q_\infty e^{-i\alpha}) + i\rho q_\infty \delta\Gamma_0 + O(\varepsilon^2) \end{aligned} \tag{14}$$

where the superscripts stand for source (1) and vortex (2), respectively, and only the leading terms in the singularity strength $\varepsilon$ are kept, which is appropriate since we are seeking the linearized force induced by the point perturbations. In Equation (14), the first term on the right-hand side corresponds to the force exerted by a source or a vortex on a body as given by Lagally's theorem [24], while the term $i\rho q_\infty \delta\Gamma_0$ reflects the contribution to the force due to the perturbation to the circulation caused by the singularities. The value of $\delta\Gamma_0$ is obtained by considering the perturbation to the Kutta condition, which fixes the value of the circulation around the body. For an airfoil with a sharp trailing edge, the point perturbations disturb the flow at the trailing edge, and the circulation has to be readjusted accordingly to prevent a flow singularity appearing at the trailing edge. For blunt bodies without sharp trailing edges, a Kutta-like condition is also required in order to derive

consistent adjoint solutions, the condition, in that case, being that the perturbation induced by the point singularity does not change the position of the rear stagnation point.

The computation of $\delta\Gamma_0$ is as follows. After inserting a point perturbation at a point $\zeta_s$ in the $\zeta$-plane, the potential receives an extra contribution from the singularity and its images [24], which have to be inserted in order to preserve the non-transpiration boundary condition. The complete potential describing the base flow + the singularity in the $\zeta$-plane is now $\Phi(\zeta) + \Phi_s(\zeta)$, where $\Phi(\zeta)$ is given in Equation (10) and

$$\Phi_s(\zeta) = \frac{\nu}{2\pi\rho}\ln(\zeta - \zeta_s) + \frac{\overline{\nu}}{2\pi\rho}\ln\left(\zeta - \frac{R^2}{\overline{\zeta}_s}\right) - \frac{\overline{\nu}}{2\pi\rho}\ln\zeta \tag{15}$$

where $\nu = \varepsilon$ for a source and $\nu = -i\varepsilon$ for a point vortex, and complex conjugation is denoted with an overbar. The perturbation potential $\Phi_s(\zeta)$ gives rise to a non-zero velocity at the trailing edge $\zeta_{te} = R$,

$$W_s(\zeta_{te}) = \left.\frac{d\Phi_s}{d\zeta}\right|_{\zeta = R} = \frac{\nu}{2\pi\rho(R - \zeta_s)} + \frac{\overline{\nu}}{2\pi\rho}\frac{1}{R - \frac{R^2}{\overline{\zeta}_s}} - \frac{\overline{\nu}}{2\pi\rho R} \tag{16}$$

which has to be compensated with the additional circulation

$$\delta\Gamma_0 = -2\pi i R W_s(\zeta_{te}) \tag{17}$$

in order to preserve the Kutta condition so that Equation (14) yields, after some rearrangement,

$$\begin{aligned}(\delta D - i\delta L)^{(1)} &= -\varepsilon q_\infty\left(\frac{R}{\zeta_s - R} - \frac{R}{\overline{\zeta}_s - R}\right) + \varepsilon e^{i\alpha}(u - iv - q_\infty e^{-i\alpha}) + O(\varepsilon^2)\\ (\delta D - i\delta L)^{(2)} &= -i\varepsilon q_\infty\left(\frac{R}{\zeta_s - R} + \frac{R}{\overline{\zeta}_s - R}\right) + i\varepsilon e^{i\alpha}(u - iv - q_\infty e^{-i\alpha}) + O(\varepsilon^2)\end{aligned} \tag{18}$$

Hence, the properly normalized linearized functionals are (we separate the real and imaginary parts in Equation (18), set $\varepsilon = 1$ and divide by $c_\infty$)

$$\begin{aligned}\delta I_D^{(1)}(\vec{\xi}) &= \tfrac{1}{c_\infty}(u\cos\alpha + v\sin\alpha - q_\infty)_{\vec{x} = \vec{\xi}}\\ \delta I_L^{(1)}(\vec{\xi}) &= \tfrac{1}{c_\infty}(v\cos\alpha - u\sin\alpha)_{\vec{x} = \vec{\xi}} + \tfrac{q_\infty}{c_\infty}\Upsilon^{(1)}(\vec{\xi})\\ \delta I_D^{(2)}(\vec{\xi}) &= \tfrac{1}{c_\infty}(v\cos\alpha - u\sin\alpha)_{\vec{x} = \vec{\xi}}\\ \delta I_L^{(2)}(\vec{\xi}) &= -\tfrac{1}{c_\infty}(u\cos\alpha + v\sin\alpha - q_\infty)_{\vec{x} = \vec{\xi}} + \tfrac{q_\infty}{c_\infty}\Upsilon^{(2)}(\vec{\xi})\end{aligned} \tag{19}$$

where

$$\begin{aligned}\Upsilon^{(1)}(z) &= -i\left(\frac{R}{\zeta(z) - R} - \frac{R}{\overline{\zeta}(z) - R}\right) = -\frac{2RY}{(X - R)^2 + Y^2}\\ \Upsilon^{(2)}(z) &= \frac{R}{\zeta(z) - R} + \frac{R}{\overline{\zeta}(z) - R} = 2\frac{R(X - R)}{(X - R)^2 + Y^2}\end{aligned} \tag{20}$$

are the contribution to lift of the circulation required to restore the Kutta condition for the source and the vortex, respectively. Recall that capital letters $(X, Y)$ denote coordinates in the $\zeta$-plane, $\zeta = X + iY$, and are given in terms of $z = x + iy$ by the inverse conformal mapping. We see from (20) that both $\Upsilon^{(1)}$ and $\Upsilon^{(2)}$ are singular at the trailing edge $\zeta_{te} = R$. Hence, the perturbation to drag is smooth in both cases, while the perturbation to lift has a singularity at the trailing edge due to the perturbations to the Kutta condition.

Additionally, $\Upsilon^{(2)}$ has a very interesting behavior on the surface of the airfoil that has very profound consequences. One can check from Equation (20) that $\Upsilon^{(2)} = -1$ throughout

the airfoil (defined by $\zeta = Re^{i\theta}$, $0 \le \theta < 2\pi$ on the $\zeta$-plane). Using this property, we find that, in the limit as the vortex approaches the surface of the airfoil, the linearized lift is

$$\delta I_L^{(2)} \rightarrow -\frac{1}{c_\infty}(u\cos\alpha + v\sin\alpha) \tag{21}$$

Since $\delta I_L^{(2)} = \psi^T f^{(2)} = (\psi_x, \psi_y) \cdot (v, -u)$ and $(v, -u) \sim q\vec{n}_S$ at the wall, it follows, therefore, from Equation (21) that

$$(\psi_x, \psi_y) \cdot \vec{n}_S = \frac{1}{c_\infty}\vec{n}_S \cdot (-\sin\alpha, \cos\alpha) \tag{22}$$

which is the adjoint wall b.c. (4). This link between the behavior of $\delta I_L^{(2)}$ near the wall and the adjoint b.c. was pointed out in [5] and constitutes a serious check on the validity of the whole approach. In fact, without $\Upsilon^{(2)}$, or if $\Upsilon^{(2)} \neq -1$ on the airfoil, the adjoint solution obtained from the Green's functions would fail to obey the wall boundary condition, which is the ultimate reason why a Kutta condition on the perturbed flow is required even on blunt bodies.

- Change in total pressure at fixed static pressure and flow direction

As in [5], the third Green's function is taken to be the response to a stagnation pressure perturbation. The source vector is $f^{(3)}(\vec{\zeta}) = q^{-2}(1, 2u, 2v)^T$, and the perturbation that it produces to either lift or drag is [7]

$$\delta I^{(3)}(\vec{\zeta}) = -\int_0^\infty ds\,\partial_s\left(q^{-2}(\vec{x}(s))\right)\delta I^{(1)}(\vec{x}(s)) + 2\int_0^\infty ds\,q^{-2}\partial_s\left(\phi(\vec{x}(s))\right)\delta I^{(2)}(\vec{x}(s)) \tag{23}$$

where $\phi$ is the local flow angle and $q^2 = u^2 + v^2$. Equation (23) involves an integration along the local streamline passing through $\vec{\zeta}$ and $s$ is the distance along the streamline downstream of $\vec{\zeta}$. Substituting Equations (19) into (23) and carrying out the integrals yields the linearized drag and lift corresponding to the third point perturbation

$$\begin{aligned}\delta I_D^{(3)}(\vec{\zeta}) &= -\frac{1}{q^2 q_\infty c_\infty}(\vec{v} - \vec{v}_\infty)^2 \\ \delta I_L^{(3)}(\vec{\zeta}) &= -\frac{2}{q^2 c_\infty}(u\sin\alpha - v\cos\alpha)_{\vec{x}=\vec{\zeta}} - \frac{q_\infty}{c_\infty}\Xi(\vec{\zeta})\end{aligned} \tag{24}$$

where

$$\Xi(\vec{\zeta}) = -\int_0^\infty ds\,\partial_s\left(q^{-2}(\vec{x}(s))\right)\Upsilon^{(1)}(\vec{x}(s)) + 2\int_0^\infty ds\,q^{-2}\partial_s\left(\phi(\vec{x}(s))\right)(1 + \Upsilon^{(2)}(\vec{x}(s))) \tag{25}$$

Again, the perturbation to drag is smooth, while the perturbation to lift (25) diverges along the dividing streamline upstream of the trailing edge, as will be discussed shortly. This includes the well-known singularity of the incoming stagnation streamline but also a new singularity along the wall.

The analytic adjoint solutions can now be computed from Equations (8), (19) and (24) as

$$\begin{pmatrix}\psi_1 \\ \psi_x \\ \psi_y\end{pmatrix} = \begin{pmatrix}1 & 0 & q^{-2} \\ u & v & 2q^{-2}u \\ v & -u & 2q^{-2}v\end{pmatrix}^{-T}\begin{pmatrix}\delta I^{(1)} \\ \delta I^{(2)} \\ \delta I^{(3)}\end{pmatrix} = \begin{pmatrix}2 & 0 & -q^2 \\ -q^{-2}u & q^{-2}v & u \\ -q^{-2}v & -q^{-2}u & v\end{pmatrix}\begin{pmatrix}\delta I^{(1)} \\ \delta I^{(2)} \\ \delta I^{(3)}\end{pmatrix} \tag{26}$$

This yields for drag the solution

$$\begin{pmatrix} \psi_1 \\ \psi_x \\ \psi_y \end{pmatrix}_{Drag} = \frac{1}{c_\infty q_\infty} \begin{pmatrix} q^2 - q_\infty{}^2 \\ q_\infty \cos \alpha - u \\ q_\infty \sin \alpha - v \end{pmatrix} \tag{27}$$

which is smooth everywhere (and simply expressible in terms of local values of the flow variables) and is identical to the analytic drag adjoint solution found in [25]. It can be readily checked that (27) obeys the adjoint wall boundary condition and the adjoint equations and compares beautifully with a numerical solution obtained with the SU2 solver (Figure 5).

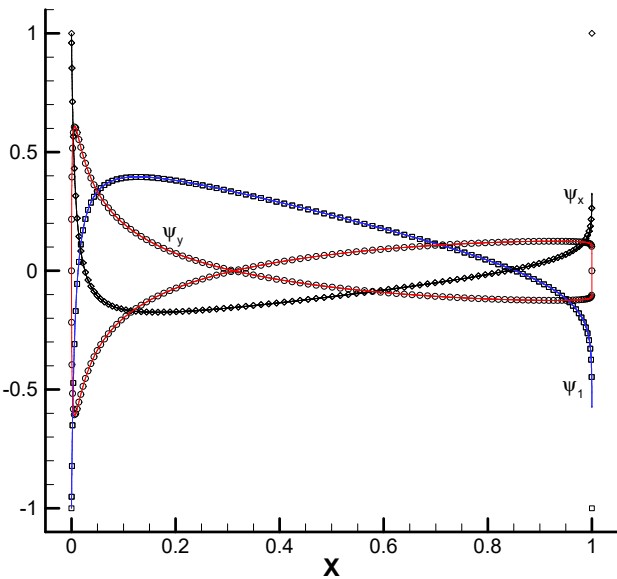

**Figure 5.** Analytic (symbols) vs. numerical (solid lines) drag–based adjoint solution on the airfoil profile for incompressible, inviscid flow at $\alpha = 0°$ past a van de Vooren airfoil with trailing–edge angle $\tau = 16°$ and 12% thickness. The numerical solution has been computed with the SU2 solver on the finest mesh of Figure 4.

For lift, Equation (26) yields the solution

$$\begin{pmatrix} \psi_1 \\ \psi_x \\ \psi_y \end{pmatrix}_{Lift} = \frac{q_\infty}{c_\infty} \begin{pmatrix} 2\Upsilon^{(1)} - q^2 \Xi \\ -q_\infty{}^{-1} \sin \alpha + u\Xi - \frac{u}{q^2}\Upsilon^{(1)} + \frac{v}{q^2}\left(1 + \Upsilon^{(2)}\right) \\ q_\infty{}^{-1} \cos \alpha + v\Xi - \frac{v}{q^2}\Upsilon^{(1)} - \frac{u}{q^2}\left(1 + \Upsilon^{(2)}\right) \end{pmatrix} \tag{28}$$

Again, it can be checked that Equation (28) obeys the adjoint wall b.c. and the adjoint equations. We see from (28) that the singularities of the linearized lift described above are transferred to the lift-based adjoint solution. The solution, therefore, has a primary singularity at the trailing edge caused by $\Upsilon^{(1)}$ and $\Upsilon^{(2)}$ and, thus, by the Kutta condition. It also has a singularity along the dividing streamline upstream of the trailing edge caused by the streamline integral $\Xi$. Recall that the value of $\Xi$ at a point $\vec{\zeta}$ in the domain is given by Equation (25), which is a downstream integration along the local streamline passing through $\vec{\zeta}$. As $\vec{\zeta}$ approaches either the incoming stagnation streamline or the wall, the local streamline approaches the singularity at the trailing edge and $\Xi$ diverges due to the divergence at the trailing edge. The trailing edge divergence thus explains both the singularities at the incoming stagnation streamline and the wall, which, in fact, show an identical behavior $\Xi \sim 1/d^{1/2+\tau/\pi}$ with the distance $d$ to the stagnation streamline or the wall. This behavior is not universal since it depends on the trailing edge angle $\tau$, and reduces to the inverse square-root behavior predicted in [5] for cusped trailing edges.

Downstream of the trailing edge, the dividing streamline is not singular (the streamline integral behaves as $\Xi \sim d$, where $d$ is the distance to the dividing streamline) except at the trailing edge itself, where $\Xi \sim 1/d^{\frac{1+2\tau/\pi}{2-\tau/\pi}}$ and $d$ is now the minimum distance to the trailing edge along the streamline. Setting $\tau = \pi$ correctly reproduces the results for blunt bodies such as the circle, and Equation (28) also applies to those cases (recall the above discussion concerning the perturbed Kutta condition for blunt bodies).

In order to produce a sample analytic solution for the case at hand, the streamline integral $\Xi$ needs to be computed. This requires prior determination of the streamline passing by a given point, which is done by direct numerical integration of the equation $d\vec{x}/dt = \vec{v}(\vec{x})$ with a fourth-order Runge–Kutta method (see e.g., [26]). The streamline tracing is performed in the circle plane and then transferred to the airfoil plane via the conformal transformation (1). The analytic lift adjoint solution obtained in this way is shown in Figure 6.

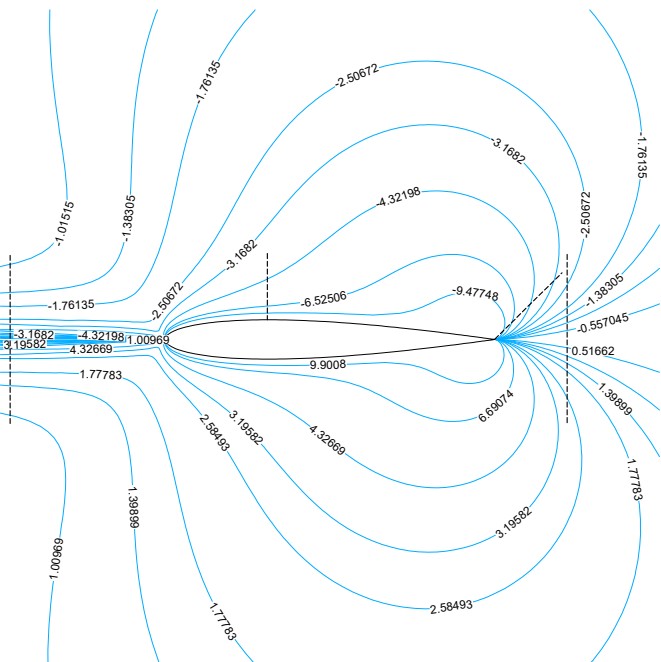

**Figure 6.** Contour map of the first component $\psi_1$ of the analytic lift–based adjoint solution for inviscid incompressible flow at $\alpha = 0°$ past a van de Vooren airfoil with trailing−edge angle $\tau = 16°$ and 12% thickness.

As expected, the solution shows singularities at the wall, the incoming stagnation streamline, and the trailing edge, but not at the rear stagnation streamline. This is more clearly illustrated in Figures 7–10, which plot the first adjoint variable $\psi_1$ along lines approaching the stagnation streamline upstream of the airfoil, the wall, the trailing edge, and the rear stagnation streamline, respectively, as indicated in Figure 6. In these plots, the numerical results obtained with the SU2 solver are also included. The analytic and numerical solutions show an excellent agreement, and both diverge as the wall is approached. It is then clear that the anomaly observed in numerical computations is caused by the divergence of the analytic solution at the wall. This is further illustrated in Figure 11, where the analytic lift adjoint solution is shown along a succession of O-shaped curves surrounding the van der Vooren airfoil profile and progressively closer to it (the O-curves are built as circumferences concentric with the circle in the circle plane and are subsequently transferred to the airfoil plane via the conformal transformation.) The analytic solution grows unbounded as the curves approach the wall. This behavior is identical to the behavior observed in [6] with the cell-centered ELSA solver, which does not directly compute

the solution at the wall. On the other hand, solvers such as SU2 and Tau use cell-vertex schemes that compute the solution at the wall. Even though the analytic solution is infinite at the wall, the numerical dissipation of the solver stabilizes the divergence, producing a finite value at the profile, which nevertheless varies continually as the grid spacing (see [1] and Figure 4) or the intensity of the numerical dissipation [4] change.

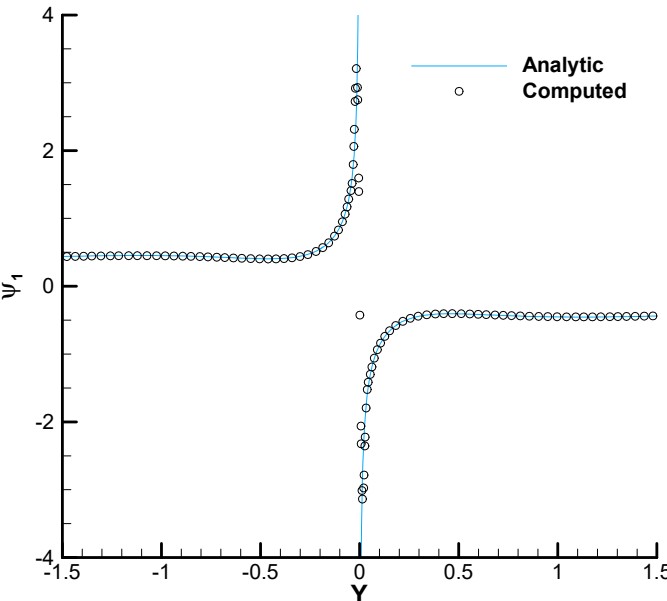

**Figure 7.** Analytic vs. numerical lift–based adjoint variable $\psi_1$ along a line crossing the stagnation streamline upstream of the airfoil as indicated in Figure 6 for incompressible, inviscid flow at $\alpha = 0°$ past a van de Vooren airfoil with trailing–edge angle $\tau = 16°$ and 12% thickness. The numerical solution has been computed with the SU2 solver on the finest mesh of Figure 4.

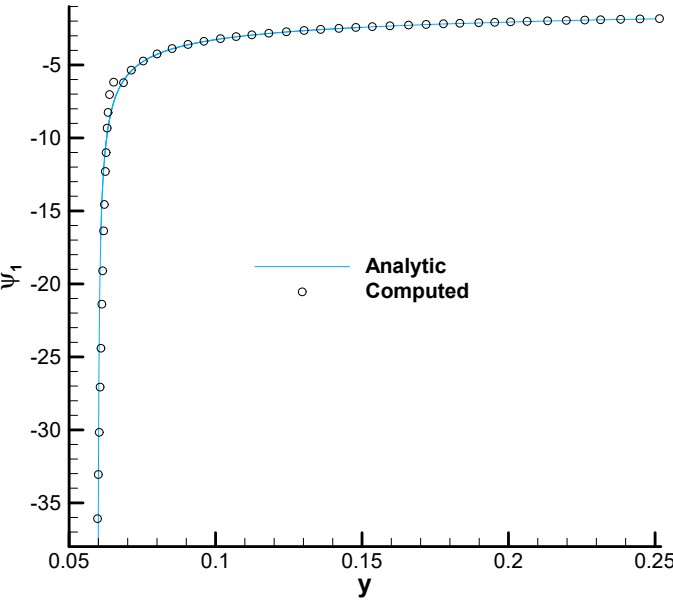

**Figure 8.** Analytic vs. numerical lift–based adjoint variable $\psi_1$ along a line normal to the airfoil wall at $x/c = 0.31$ as indicated in Figure 6 for incompressible, inviscid flow at $\alpha = 0°$ past a van de Vooren airfoil with trailing–edge angle $\tau = 16°$ and 12% thickness. The numerical solution has been computed with the SU2 solver on the finest mesh of Figure 4. The wall is at $y = 0.06$.

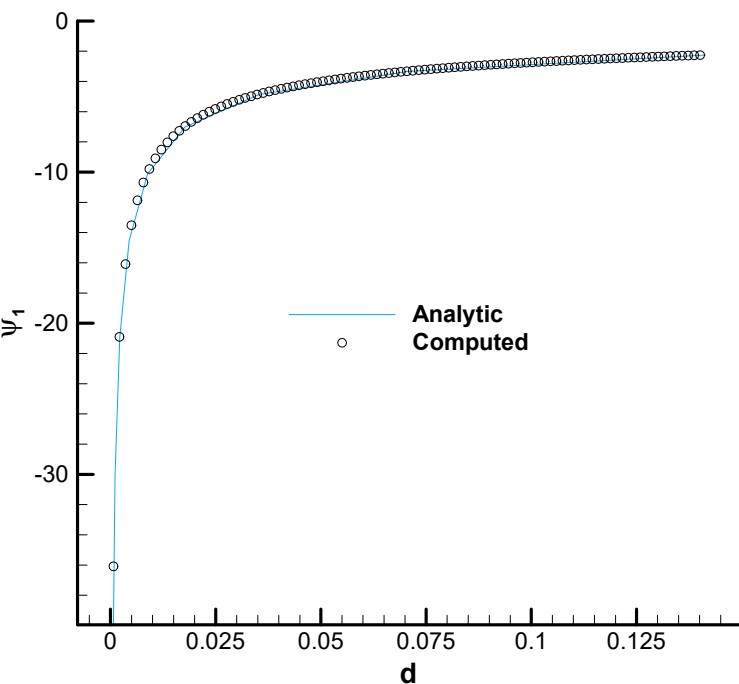

**Figure 9.** Analytic vs. numerical lift–based adjoint variable $\psi_1$ along the line $(x, y) = (1 + d/\sqrt{2}, d/\sqrt{2})$ approaching the trailing edge as indicated in Figure 6 for incompressible, inviscid flow at $\alpha = 0°$ past a van de Vooren airfoil with trailing–edge angle $\tau = 16°$ and 12% thickness. The numerical solution has been computed with the SU2 solver on the finest mesh of Figure 4. $d$ denotes the distance to the trailing edge.

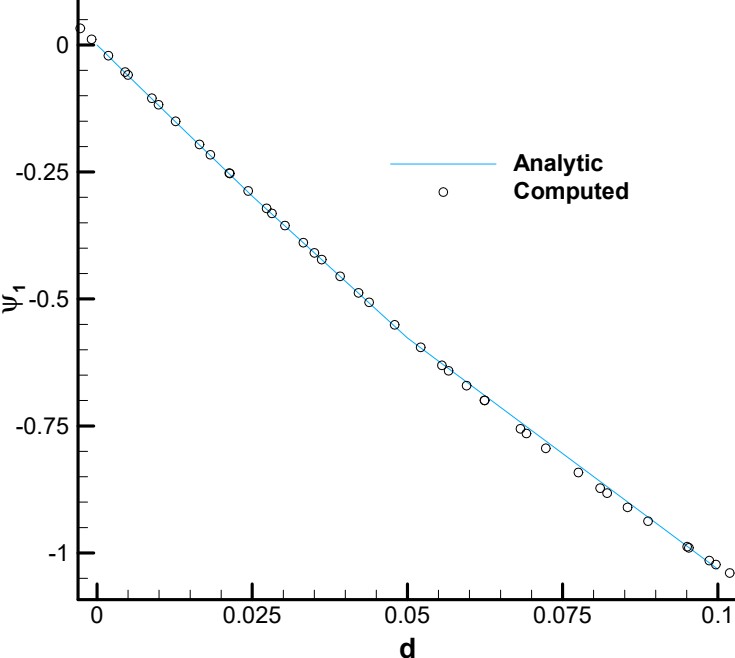

**Figure 10.** Analytic vs. numerical lift–based adjoint variable $\psi_1$ along a line crossing the stagnation streamline downstram of the airfoil as indicated in Figure 6 for incompressible, inviscid flow at $\alpha = 0°$ past a van de Vooren airfoil with trailing–edge angle $\tau = 16°$ and 12% thickness. The numerical solution has been computed with the SU2 solver on the finest mesh of Figure 4. $d$ denotes the distance to the stagnation streamline.

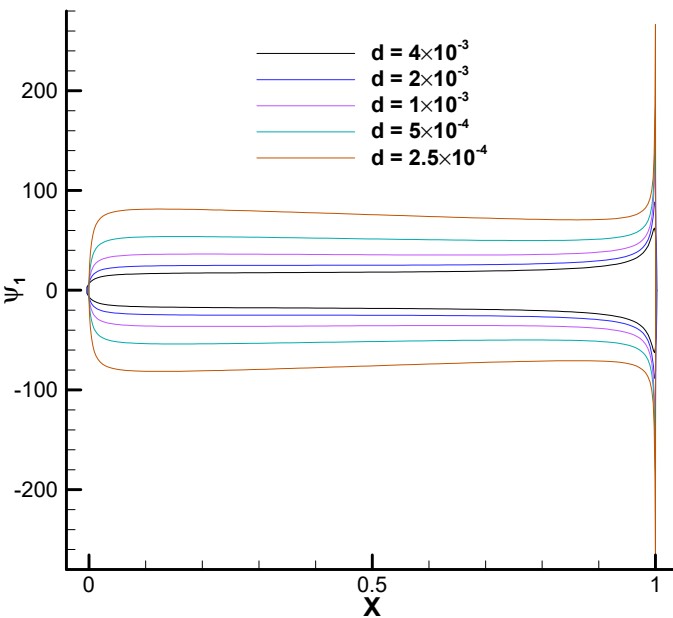

**Figure 11.** Analytic lift–based adjoint solution for incompressible, inviscid flow at $\alpha = 0°$ past a van de Vooren airfoil with trailing–edge angle $\tau = 16°$ and 12% thickness on a sequence of O–shaped curves surrounding the airfoil at decreasing distance $d$.

Finally, Figures 12 and 13 how the value of the linearized lift functionals $\delta I_L^{(1)}$ and $\delta I_L^{(2)}$ on the surface of the airfoil computed with the analytic solution and the numerical adjoint solution obtained with the SU2 solver on the sequence of meshes of Figure 4. Notice that $\delta I_L^{(1)}$ (the linearized perturbation to the lift caused by a point mass source) is related to the (continuous) adjoint-based lift gradient [12]

$$\delta \int_S C_p \left( \vec{n}_S \cdot \vec{d} \right) ds = \int_S c_\infty^{-1} (\delta \vec{x} \cdot \vec{n}_S)(\vec{d} \cdot \nabla p) ds - \int_S (\vec{n}_S \cdot \delta \vec{v}) \rho (\psi_1 + \vec{v} \cdot (\psi_x, \psi_y)) ds \tag{29}$$

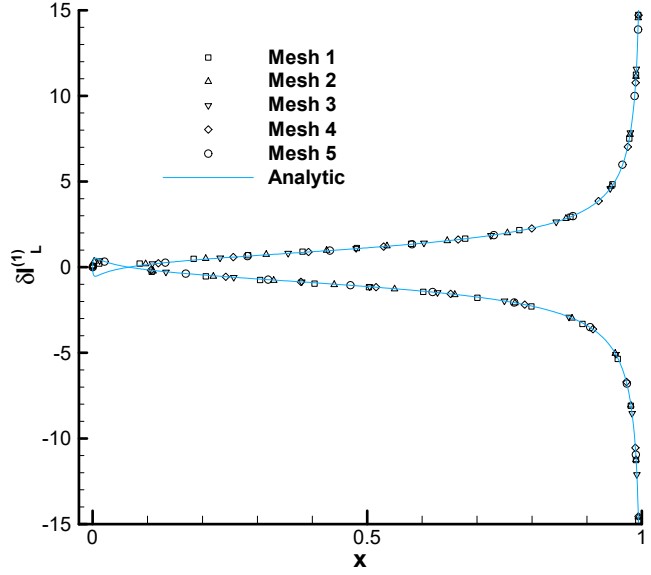

**Figure 12.** $\delta I^{(1)} = \psi_1 + \vec{v} \cdot (\psi_x, \psi_y)$ computed with numerical and analytic lift–based adjoint solutions for incompressible, inviscid flow at $\alpha = 0°$ past a van de Vooren airfoil with trailing–edge angle $\tau = 16°$ and 12% thickness.

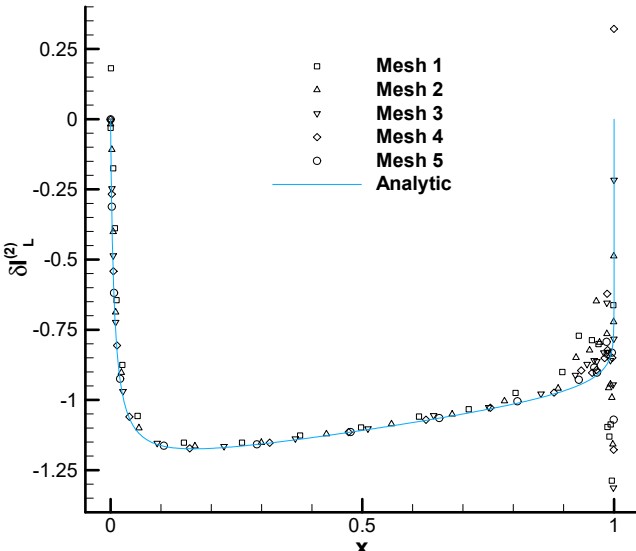

**Figure 13.** $\delta I^{(2)} = v\psi_x - u\psi_y$ computed with numerical and analytic lift–based adjoint solutions for incompressible, inviscid flow at $\alpha = 0°$ past a van de Vooren airfoil with trailing–edge angle $\tau = 16°$ and 12% thickness.

($\delta\vec{v}$ is the perturbed velocity, and the first term on the right-hand side stands for the purely geometric variation of the cost function which need not concern us here), while $\delta I_L^{(2)}$ (the linearized perturbation to the lift caused by a point vortex) approaches $q\vec{n}_S \cdot (\psi_x, \psi_y)$ as the point approaches the wall and is thus directly related to the adjoint b.c.

We note three significant points here:

- There is a nice agreement between the analytic and numerical results. It is clear (again) that the analytic solution can be used for verification of numerical adjoint solvers.
- In the previous point, we probably overlooked the fact that the quantities computed with the analytic solution are finite in spite of the fact that the analytic solution diverges at the wall.
- Similarly, the quantities computed with the numerical solution are stable against mesh refinement despite being computed with an adjoint solution that diverges with mesh refinement.
- The explanation of the above facts is simple in view of the analytic solution (28). As was already conjectured in [4], the adjoint variables $\psi_1, \psi_x, \psi_y$ diverge towards the wall while the combinations

$$\delta I_L^{(1)} = \psi_1 + \vec{v} \cdot (\psi_x, \psi_y) = (q_\infty \Upsilon^{(1)} - u \sin\alpha + v\cos\alpha)/c_\infty$$
$$\delta I_L^{(2)} = v\psi_x - u\psi_y = (-v \sin\alpha - u\cos\alpha + q_\infty(1 + \Upsilon^{(2)}))/c_\infty \tag{30}$$

remain finite except at the trailing edge. This is a serious check of the validity of the solution and explains why a divergent adjoint solution can obey finite boundary conditions and lead to well-defined sensitivities.

## 4. Analytic Adjoint Solution for Subcritical Flow

Numerical adjoint solutions for subcritical flows are very similar to their incompressible counterparts. Drag-based adjoint solutions are free of singularities and converge well with mesh refinement, while lift-based adjoint solutions possess singularities along the dividing streamline upstream of the trailing edge and diverge with mesh refinement, as illustrated in Figure 14, where the similarity to the incompressible case (Figure 4) is evident.

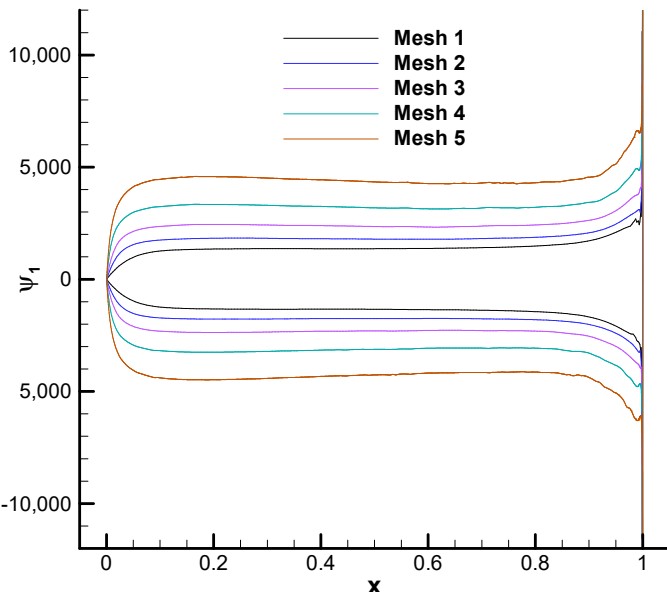

**Figure 14.** Lift–based adjoint solution on the van de Vooren airfoil profile at subcritical flow conditions $M_\infty = 0.5$ and $\alpha = 0°$ computed with the SU2 solver on 5 progressively refined unstructured triangular meshes.

In order to extend the preceding analysis to this case, analytic flow and adjoint solutions are required. Unfortunately, no analytic flow solution is known in general for the subcritical case, and no systematic procedure for building exact Green's functions is known to the authors. However, we do not need the full Green's functions but only their corresponding linearized functionals. In the incompressible case, we obtained them using Lagally's theorem, which can be demonstrated by transferring the integral over the wall to the farfield boundary using the asymptotic forms of the flow near the singularity and the farfield [24]. We could try to explore the possibility of using the same approach in the compressible case using the linearized flow Equation (6) and making assumptions about the asymptotic structure of the perturbed flow at the farfield following [27,28], but we have not attempted this approach here. Instead, we have followed a more direct, somewhat heuristic route that takes advantage of the fact that an analytic adjoint solution for the drag-based adjoint equations that is valid for two-dimensional inviscid isentropic irrotational flows (such as, for example, subcritical inviscid flow past 2D airfoils) has been obtained in [25]. The solution is the following

$$
\psi_D = \frac{1}{\rho_\infty^{1-\gamma} q_\infty c_\infty}
\begin{pmatrix}
H\rho^{1-\gamma} - H_\infty \rho_\infty^{1-\gamma} \\
-\rho^{1-\gamma} u + \rho_\infty^{1-\gamma} q_\infty \cos\alpha \\
-\rho^{1-\gamma} v + \rho_\infty^{1-\gamma} q_\infty \sin\alpha \\
\rho^{1-\gamma} - \rho_\infty^{1-\gamma}
\end{pmatrix}
\tag{31}
$$

where $H$ is the enthalpy and $\gamma$ is the ratio of specific heats. Using this solution, it is possible to compute the linearized drag functionals for the four point perturbations introduced in [5] (mass source, point force normal to the local flow, and point perturbations to the stagnation enthalpy and pressure) as

$$
\delta I_D =
\begin{pmatrix}
\delta I_D^{(1)} \\
\delta I_D^{(2)} \\
\delta I_D^{(3)} \\
\delta I_D^{(4)}
\end{pmatrix}
=
\begin{pmatrix}
1 & 0 & -\frac{1}{2H} & \frac{1}{p_0}\left(\frac{\gamma-1}{\gamma} + \frac{1}{\gamma M^2}\right) \\
u & -\rho v & 0 & \frac{u}{p_0}\left(\frac{\gamma-1}{\gamma} + \frac{2}{\gamma M^2}\right) \\
v & \rho u & 0 & \frac{v}{p_0}\left(\frac{\gamma-1}{\gamma} + \frac{2}{\gamma M^2}\right) \\
H & 0 & \frac{1}{2} & \frac{H}{p_0}\left(\frac{\gamma-1}{\gamma} + \frac{1}{\gamma M^2}\right)
\end{pmatrix}^T
\psi_D = \frac{1}{c_\infty q_\infty}
\begin{pmatrix}
(\vec{q} - \vec{q}_\infty) \cdot \vec{q}_\infty \\
\rho(-v, u) \cdot \vec{q}_\infty \\
0 \\
-\frac{1}{\rho_0 q^2}(\vec{q} - \vec{q}_\infty)^2
\end{pmatrix}
\tag{32}
$$

where $\rho_0 = \frac{\gamma}{\gamma-1}\frac{p_0}{H}$ is the (constant) stagnation density. In view of Equation (32), and by analogy with the incompressible case, we can infer that the forces exerted by the "source" and the "vortex" (the first two perturbations) on the wall are, respectively,

$$
\begin{aligned}
\vec{F}_1 &= \vec{q} - \vec{q}_\infty \\
\vec{F}_2 &= \rho(\vec{q} - \vec{q}_\infty)^\perp
\end{aligned}
\tag{33}
$$

where $\vec{q}^\perp \equiv (-v, u)$. Notice that these agree with the first two rows in Equation (32) if we project the force vectors along $\hat{q}_\infty = (\cos\alpha, \sin\alpha)$ and divide by $c_\infty$. Any additional component along $\hat{q}_\infty^\perp = (-\sin\alpha, \cos\alpha)$ can be discarded as far as the lift adjoint ansatz presented in Equation (35) below is concerned, as it can be absorbed into the Kutta functions.

As for the fourth row in Equation (32), it is related to the first two, as in the incompressible case, by means of a streamline integral, which for compressible flows takes the form

$$
\delta I^{(4)} = -\frac{1}{\rho_0}\left(\int_0^\infty ds' \partial_s q^{-2}\delta I^{(1)} + \int_0^\infty ds' \frac{2}{\rho q^2}\partial_s \phi \delta I^{(2)}\right)
\tag{34}
$$

It is easy to check that, substituting $\delta I_D^{(1)}$ and $\delta I_D^{(2)}$ from Equations (32) into (34) one recovers $\delta I_D^{(4)}$. There are some minor differences between the above formulae and their incompressible counterparts that stem from the different normalization of the source terms in [5,7].

We can now make an ansatz for the linearized lift functionals using Equations (33) and (34). For lift, the forces are projected along $\hat{q}_\infty^\perp = (-\sin\alpha, \cos\alpha)$. A second, a priori singular part needs to be added. This part contains the lift generated by the additional circulation required to maintain smooth flow at the trailing edge in the presence of the point perturbations, and we make the assumption that it can be incorporated as in Equation (19) for the incompressible case. This procedure results in the following ansatz for the linearized lift

$$
\begin{aligned}
\delta I_L^{(1)} &= \vec{F}_1 \cdot \hat{q}_\infty^\perp + Kutta_1 = \frac{1}{c_\infty q_\infty}(vu_\infty - uv_\infty) + \frac{q_\infty}{c_\infty}\Upsilon^{(1)} \\
\delta I_L^{(2)} &= \vec{F}_2 \cdot \hat{q}_\infty^\perp + Kutta_2 = \frac{\rho}{c_\infty q_\infty}\left(\vec{q} - \vec{q}_\infty\right)\cdot \vec{q}_\infty - \rho\frac{q_\infty}{c_\infty}\Upsilon^{(2)} \\
\delta I_L^{(3)} &= 0 \\
\delta I_L^{(4)} &= -\frac{1}{\rho_0}\left(\int_0^\infty ds'(\partial_s q^{-2}\delta I^{(1)} + \frac{2}{\rho q^2}\partial_s \phi \delta I^{(2)})\right) \\
&= -\frac{2}{\rho_0 c_\infty q_\infty q^2}(uv_\infty - vu_\infty) + \frac{q_\infty}{\rho_0 c_\infty}\Xi
\end{aligned}
\tag{35}
$$

where $\Upsilon^{(1)}$ and $\Upsilon^{(2)}$ are two unknown functions and

$$
\Xi = -\int_0^\infty ds'(\partial_s q^{-2}\Upsilon^{(1)} - \frac{2}{q^2}\partial_s \phi(1 + \Upsilon^{(2)}))
\tag{36}
$$

is the streamline integral for the compressible case. Using Equation (35) and the inverse of Equation (32), we get the following form for the lift-based analytic adjoint solution for subcritical flows

$$
\begin{pmatrix}\psi_1 \\ \psi_x \\ \psi_y \\ \psi_4\end{pmatrix}_{Lift} = \frac{q_\infty}{c_\infty}\begin{pmatrix}\frac{(\gamma-1)\rho}{2\gamma p}(2\Upsilon^{(1)} - q^2\Xi)H \\ -\frac{\sin\alpha}{q_\infty} + \frac{(\gamma-1)\rho H}{\gamma p}u\Xi - \frac{(\gamma-1)\rho}{\gamma p}(H + \frac{1}{2}q^2)\frac{u}{q^2}\Upsilon^{(1)} + \frac{v}{q^2}\left(1 + \Upsilon^{(2)}\right) \\ \frac{\cos\alpha}{q_\infty} + \frac{(\gamma-1)\rho H}{\gamma p}v\Xi - \frac{(\gamma-1)\rho}{\gamma p}(H + \frac{1}{2}q^2)\frac{v}{q^2}\Upsilon^{(1)} - \frac{u}{q^2}\left(1 + \Upsilon^{(2)}\right) \\ \frac{(\gamma-1)\rho}{2\gamma p}(2\Upsilon^{(1)} - q^2\Xi)\end{pmatrix}
\tag{37}
$$

The solution (37) is only valid for steady two-dimensional inviscid isentropic irrotational flows. Unlike in the incompressible case, the form of the Kutta functions $\Upsilon^{(1)}$ and $\Upsilon^{(2)}$ in Equations (35)−(37) which are the contributions to lift of the trailing edge condition for the first two point perturbations, is a priori unknown and may be very hard to guess. However, the adjoint b.c. (4) requires that $\Upsilon^{(2)} = -1$ at the airfoil profile. Likewise, demanding that Equation (37) obeys the adjoint equations puts the following additional constraints on $\Upsilon^{(1)}$ and $\Upsilon^{(2)}$

$$
\begin{aligned}
(M^2 - 1)\rho \vec{v} \cdot \nabla \Upsilon^{(1)} + (v, -u) \cdot \nabla(\rho(1 + \Upsilon^{(2)})) &= 0 \\
\rho(v, -u) \cdot \nabla \Upsilon^{(1)} + \vec{v} \cdot \nabla(\rho(1 + \Upsilon^{(2)})) &= 0
\end{aligned}
\tag{38}
$$

where $M$ is the local Mach number. In the incompressible case, $\Upsilon^{(1)}$ and $\Upsilon^{(2)}$ are the imaginary and real parts of a meromorphic function and, thus, obey the Cauchy–Riemann equations $\partial_x \Upsilon^{(1)} = -\partial_y \Upsilon^{(2)}$ and $\partial_y \Upsilon^{(1)} = \partial_x \Upsilon^{(2)}$. Multiplying alternatively $\nabla \Upsilon^{(1)}$ and $\nabla \Upsilon^{(2)}$ by the velocity vector $\vec{v}$ and using the Cauchy–Riemann equations leads to Equation (38) with constant $\rho$ and $M = 0$.

Using streamline coordinates, Equation (38) can be written as

$$
\begin{aligned}
(1 - M^2)\rho \partial_s \Upsilon^{(1)} + \partial_n(\rho(1 + \Upsilon^{(2)})) &= 0 \\
\rho \partial_n \Upsilon^{(1)} - \partial_s(\rho(1 + \Upsilon^{(2)})) &= 0
\end{aligned}
\tag{39}
$$

where $s$ is the coordinate along streamlines and $n$ is the coordinate perpendicular to streamlines. Since $\Upsilon^{(2)} = -1$ at the wall, it follows from the second equation in Equation (39) that $\partial_n \Upsilon^{(1)}\big|_{wall} = 0$.

It is clear from Equation (37) and the subsequent arguments that the structure of the solution is very similar to the incompressible case. The numerical solutions are also quite similar and have the same set of singularities, even though the precise values of the adjoint variables differ. The solution given in Equation (37) will have singularities at the trailing edge and the dividing streamline upstream of the trailing edge if $\Upsilon^{(1)}$ and $\Upsilon^{(2)}$ are singular at the trailing edge and grow sufficiently fast towards it. That $\Upsilon^{(1)}$ and $\Upsilon^{(2)}$ diverge at the trailing edge is a reasonable assumption since the closer the perturbation to the trailing edge, the greater the disturbance on the trailing edge flow and the higher the required adjustment in the circulation. The fact that they diverge with the appropriate exponent can be inferred from the numerical solution having the same set of singularities as in the incompressible case. We can thus conclude that the cause of the numerical mesh divergence observed in subcritical cases is the same as in the incompressible case.

## 5. Summary and Discussion

Direct analysis of the behavior of the analytic lift-based adjoint solution for the incompressible Euler equations in 2D shows that the adjoint solution is singular at the wall and the incoming stagnation streamline. The ultimate origin of both singularities is to be found in the adjoint singularity at the trailing edge, which is due to the sensitivity of the Kutta condition to perturbations of the flow [7,29].

Numerical adjoint solutions to the Euler equations in two and three dimensions for various flow conditions exhibit a divergent behavior with mesh refinement near solid walls. The comparison between analytic and numerical adjoint solutions for a representative incompressible case confirms that the numerical mesh divergence is due to the singularity of the analytic solution. The effect of the adjoint trailing edge singularity is local for point source and vortex perturbations but extends upstream along the dividing streamline for stagnation pressure perturbations, thus explaining the results in [5,6]. Numerical solutions computed with cell-vertex schemes, which place computational nodes directly on the geometry, do not directly show the divergence owing to numerical dissipation. The price to be paid is that the numerical solution at the wall depends continually on the level of dissipation or the mesh density. Decreasing the dissipation or refining the mesh, which

has similar effects on the solution, changes the value of the adjoint solution at the wall. A similar effect can be found in solutions computed with cell-centered schemes, only that now the continuous variation with mesh density affects the values computed at the near-wall cells.

While we have only produced an analytic solution for incompressible cases, we think that it is safe to extend the conclusions to compressible flows as well, at least qualitatively. This is particularly clear for subcritical flows, for which both the numerical solution and the structure of the analytic solution are strikingly similar to the incompressible case. For other compressible flows, the situation depends on the structure of the flow around the trailing edge. When perturbations to the trailing edge flow are suppressed by the flow conditions (high transonic, supersonic, or viscous cases, in which the structure of the flow at the trailing edge is tightly constrained) the singularities disappear, and the numerical adjoint solutions behave smoothly with mesh refinement. In other cases (at low or medium transonic speeds), mesh-diverging numerical adjoint solutions also exhibit the singularities at the wall, the trailing edge, and the incoming stagnation streamline. It is reasonable to assume that, as in the incompressible or subcritical cases, these singularities simply reflect the sensitivity of the lift or drag to perturbations to the Kutta condition. In principle, the Kutta condition affects circulation and thus lift, with one exception: Rotational transonic flow, e.g., transonic flow with a shock on the upper surface. In these cases, the total pressure loss is larger on the upper side of the trailing edge. This forces the flow to stagnate at the trailing edge upper surface while the velocity at the trailing edge lower surface is finite and non-zero. The flow thus leaves the trailing edge tangent to the lower surface (the surface with the higher total pressure) smoothly and forms a slip line [30]. In this situation, if a point perturbation disrupts the flow at the trailing edge, restoring the Kutta condition requires additional circulation but also the readjustment of the shock wave position [6], creating drag.

It would be very interesting to confirm the above arguments with a deeper understanding of the analytic solutions for transonic cases. The subcritical case is also missing a few key ingredients and any insight in that direction would be welcome, even though the lack of a general procedure for deriving even analytic flow solutions in this and transonic cases is certainly an obstacle to the derivation of closed-form analytic adjoint solutions. Future developments along these lines will lead to an improved understanding of the behavior of adjoint solutions, as the incomplete analysis presented in Sections 4 and 5 clearly exemplifies. We hope to return to these issues in the future.

**Author Contributions:** Both authors have contributed equally to the paper. All authors have read and agreed to the published version of the manuscript.

**Funding:** The research described in this paper has been supported by INTA and the Ministry of Defence of Spain under the grants Termofluidodinámica (IGB99001) and IDATEC (IGB21001).

**Institutional Review Board Statement:** Not applicable.

**Informed Consent Statement:** Not applicable.

**Data Availability Statement:** The data presented in this study are available on request from the corresponding author.

**Acknowledgments:** The numerical computations reported in the paper have been carried out with the SU2 code, an open source platform developed and maintained by the SU2 Foundation.

**Conflicts of Interest:** The authors declare no conflict of interest.

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
