# Peer review of "Explaining the Lack of Mesh Convergence of Inviscid Adjoint Solutions near Solid Walls for Subcritical Flows"

_aerospace, doi:10.3390/aerospace10050392_

Round 1

Reviewer 1 Report

The divergence of numerical solutions to the adjoint Euler equations with mesh refinement is addressed. The paper is well written and merits publication in Aerospace.

Author Response

Dear reviewer,

We would like to thank you for your kind assessment of our work. Best regards.

Reviewer 2 Report

The paper presents a proof as to why the adjoint solution to inviscid, incompressible flows and for particular objective functions (such as lift) tends to diverge as meshes become finer, despite the fact that sensitivity derivatives seem to remain unaffected by the increase of the magnitude of the adjoint fields. This paper continues the work presented by the authors in previous papers, but this time attempts to explain the above-mentioned phenomenon by deriving an analytical solution to the adjoint equations, to act as the reference for the numerical one,, something that is very interesting and adds to the novelty and the significance of the work presented herein.

The authors succeed in deriving an analytical solution, and hence explaining the behavior of the numerical one too, for incompressible, inviscid flows and attempt to follow the same line of reasoning to explain similar numerical divergence of the adjoint fields in compressible, inviscid flows without a shock. Though for the latter case, this does not seem possible without some assumptions and is more an understanding rather that a proof, this development adds to our current understanding of the phenomenon and is thus useful for the paper and the literature. 

The paper is generally well written, both language-wise and from the scientific point of view. The following minor additions might be beneficial to the final form of the paper:

1) For completeness sake, a figure similar to Fig. 3 could be included but for the primal fields, to show the flow solution indeed converges as the mesh becomes finer. 

2) p5, l142: the authors state that the adjoint wall boundary condition (b.c.) is "reasonably well obeyed across the mesh levels", while laying out investigations conducted in previous works. Why wouldn't the  b.c. not be obeyed exactly? I guess it has to do with the fact that a vertex-centered solver is used and boundary conditions are imposed on the wall in a weak sense, right? This should be mentioned explicitly, as it leaves the reader wondering. 

3) p7, l238: "By a well-known feature ....". A reference would be nice here. 

Reviewer 3 Report

This manuscript discusses the issue of numerical solutions to the adjoint Euler equations diverging with mesh refinement near walls in various flow conditions and geometries. The authors provide an explanation by comparing a numerical incompressible adjoint solution to an analytic one, revealing that the anomaly is caused by the divergence of the analytic solution at the wall. The singularity causing this divergence is related to the well-known singularity along the incoming stagnation streamline, both originating at the adjoint singularity at the trailing edge. The authors extend their argument to cover fully compressible cases in subcritical flow conditions, presenting an analytic solution with a similar structure as the incompressible one.

Strengths:

  1. The manuscript addresses an important issue in computational fluid dynamics and provides a clear explanation for the observed divergence of numerical solutions near walls.
  2. The authors present a comprehensive analysis of the problem, including various test cases and comparisons to analytic solutions.
  3. The paper is well-structured and provides a detailed background on the adjoint method and the mesh divergence problem.

Comments for improvements:

1.     The paper primarily focuses on the analysis of the analytic lift-based adjoint solution for incompressible Euler equations in 2D. While the authors do discuss extending their conclusions to compressible flows, they have not provided a comprehensive analysis or direct evidence to support this assertion. This limitation could potentially affect the applicability of the findings to a wider range of fluid dynamics problems.

  1. The authors discuss numerical adjoint solutions for the Euler equations in two and three dimensions but do not provide enough details about the specific numerical methods used or how they have addressed potential numerical issues. For example, they mention the use of cell-vertex schemes and cell-centered schemes, but do not elaborate on their implementation or any steps taken to mitigate the impact of numerical dissipation. Providing more information about the numerical methods used and any modifications made to improve their performance would enhance the manuscript's credibility.
  2. Insufficient exploration of transonic cases: The authors acknowledge that there is a need for a deeper understanding of the analytic solutions for transonic cases. However, they do not delve into this area in the current manuscript, potentially leaving gaps in the understanding of adjoint solutions in this important regime. Future research addressing these transonic cases would be beneficial for a more complete analysis of adjoint solutions in fluid dynamics problems.
